# Composting Old Bark and Wood Waste in Cold Weather Conditions

Yuliya Margina [1], Aleksandr Troegubov [1], Yuliya Kulikova [2] and Natalia Sliusar [1,*]

1   Environmental Protection Department, Perm National Research Polytechnic University, 614000 Perm, Russia;
    makarova_u85@mail.ru (Y.M.); troegubov.alexandr@mail.ru (A.T.)
2   Institute of Living Systems, Immanuel Kant Baltic Federal University, 236016 Kaliningrad, Russia;
    kulikova.pnipu@gmail.com
*   Correspondence: nnslyusar@gmail.com

**Abstract:** The pulp and paper industry generates large quantities of bark and wood waste (BWW), most of which is disposed of at bark dumps. There are dozens of such dumpsites in Russia, some of which cause negative environmental impacts due to their proximity to bodies of water. Unlike fresh BWW, old BWW is characterized by significant heterogeneity. Given that BWW stored long-term in bark dumps is a water-heavy woody material subjected to varying degrees of microbiological decomposition, the most acceptable method for its disposal is composting. This text presents the results of studies focused on the process of field composting BWW in heaps with natural aeration during the cold season in the region of Perm, Russia. Composting was carried out in two ways: (1) with mineral fertilizers; (2) with both mineral fertilizers and a microbiological inoculum. Concurrent with the field composting, laboratory composting was carried out under controlled conditions. At the end of a 60-day process of field composting old BWW at ambient temperatures of 5 to −14 °C, there were decreases in the values of the compost mixture: loss on ignition (LOI) fell by 22%, chemical oxygen demand (COD) by 98%, and respiratory activity ($AT_4$) by 32%. In laboratory conditions at an ambient temperature of 30–35 °C, LOI decreased by 24%, COD by 98%, and $AT_4$ by 39%. The introduction of a microbiological inoculum into the compost mixture did not intensify the biochemical destruction process of old BWW, neither in the laboratory nor in field conditions.

**Keywords:** biomass; bark waste; composting; lignin; wood forest product

## 1. Introduction

The Russian Federation has a well-developed timber processing industry. In the course of obtaining wood at processing enterprises, large-tonnage waste accumulates, including bark, which accounts for an average of up to 10% of the total waste volume [1–3]. Bark and wood waste (BWW) generation in Russia is at a volume of about 2 million tons per year [4], of which only 2% is processed [5], while the remainder is transported to bark dumps for storage.

For fresh BWW, there are processing technologies such as the production of wood-based panels, wood-based activated carbon, or fuel pellets. In most cases, however, wood processing companies prefer to place BWW on bark dumps, as this approach is easy to implement and imposes minimal financial costs.

The production of goods from BWW requires high-quality raw materials, as well as stable waste composition and properties to guarantee the stability of operating parameters for the production process and to ensure high-quality output. Old BWW from bark dumps does not meet these requirements as it tends to be highly heterogeneous in its properties [6].

One of the treatment methods for BWW from bark dumps is incineration. Biomass combustion is a carbon-neutral waste disposal method, but bark is a low-grade fuel with high moisture and ash content and low flow properties. Special preparation of the bark, including shredding and drying, is required prior to incineration. Financial resources are

also necessary for the purchase and operation of an incinerator, along with a dust and gas cleaning system. The low cost of electricity and thermal energy in Russia further reduces the appeal for businesses in making the effort to use heat and electricity generated from BWW incineration.

Thus, wood processing companies are taking the easiest path to gain the maximum financial benefit here and now by disposing of fresh BWW in bark dumps, thereby placing the burden on future generations to solve these problems and violating the principles of sustainable development. Some companies have gone out of business, but a large number of old bark dumps still remain.

During long-term storage in bark dumps, BWW partially decomposes with the formation of phenolic compounds and other toxic decomposition products [6]. These are washed away into the environment by precipitation and meltwater [7,8] to detrimental effect. Another risk is the occurrence of fires that are difficult to eliminate [9]. Furthermore, when wood waste decomposes under the anaerobic conditions of bark heaps, it produces the greenhouse gases $CO_2$, $CH_4$, and $N_2O$.

Since BWW that is stored long-term in bark dumps is a water-heavy woody material with a moisture content of 65–85% and is subjected to varying degrees of microbiological decomposition [10], the most acceptable method for its disposal is composting.

The process of old BWW decomposition already starts in the bark dump under anaerobic conditions and composting allows it to be completed in a shorter time frame under aerobic conditions. This makes it possible to avoid the formation of $CH_4$. Under aerobic conditions, the predominant gas formed is $CO_2$, which is of biological rather than anthropogenic origin, so its contribution to the greenhouse effect is not taken into account.

One of the main advantages of composting BWW is the possibility of obtaining finished compost that is similar to peat soils. For some areas of the Russian Federation, the possibility of replenishing soil resources is of particular interest, especially where it is common to find sod-podzolic soils with a shallow depth of the fertile layer [11].

During the composting process, a community of microorganisms from various groups [12–14] decomposes organic waste under aerobic conditions and forms carbon dioxide, water, heat, and a humus-like decomposition product [15], which, depending on the properties obtained, can be used as the technical soil or fertilizer [16,17].

The microbiological decomposition processes of the organic substrate during composting are most active at favorable ambient temperatures in the warm season [15]. At low ambient temperatures, heat loss from a heap can exceed the amount of heat generated from microbiological destruction, thereby provoking a halt in the composting process and causing the compost mass in the heap to freeze [18–20].

The storage sites for old BWW and the sources of new BWW formation are situated in territories of the Russian Federation that are covered with forest vegetation and located in the Atlantic continental climate zone. However, in temperate continental and subarctic marine climatic zones, the atmospheric temperature does not rise above +10 °C for five to eight months of the year. To minimize the negative impact of cold weather, the composting process can be carried out as in-vessel composting, which entails minimal heat exchange with the atmosphere. There are several different in-vessel composting methods using a range of equipment including containers, silos [21], agitated bays [22], tunnels [23,24], and rotating drums [25]. In-vessel composting can be high-tech, with automatic systems for monitoring composting conditions (temperature, humidity, and oxygen content) and for maintaining optimal levels using forced aeration, humidification, and mixing systems [26] along with additional floor and wall heating [23].

Thanks to such a high degree of control over the conditions, composting time can be significantly reduced, from about eight weeks to only three to eight days with the most modern systems [26]. In a recent study [27], it was found that when fresh waste from pig farms moved to the thermophilic phase within a few hours during in-vessel composting, a stable relatively high temperature (55–65 °C) was maintained, which significantly reduced the composting period and led decomposition of organic matter to reach 35.7% within the

first week. Composting plant waste and tree leaves in a rotating drum made it possible to obtain high-quality compost with a total nitrogen content of 2.6% and a final phosphorus level of 6 g/kg after a week [25].

A relatively new commercial-scale composting technology uses a combination of a GORE® Cover membrane over the compost pile and controlled aeration, thus providing protection for the composted material against wind, heat, cold, rain, and desiccation. This makes it possible to obtain mature compost in just 30 days as opposed to the standard period of 90–270 days for traditional pile composting without membrane coverage and forced aeration [28]. According to the German company UTV AG, there has been a successful year-round implementation of their GORE® Cover membrane composting system in the cities of Moncton and Saskatoon, Canada, where winter temperatures drop below −20 °C. Unfortunately, despite all their benefits, in-vessel composting systems have one significant drawback (even with the GORE® Cover membrane), namely, they have high capital costs and consequently lower economic efficiency.

Among existing technologies, field pile composting is the least expensive and easiest to implement, but it is more dependent on environmental conditions that can affect the composting process. In cold climates, composting in piles is widely considered to be impossible because the compost mass in the pile cools down and the process is stopped. To be an effective management practice, composting must be a year-round option.

In a number of research works, authors have shown in practice that the composting process inside a heap continues even when the atmospheric air is at negative temperatures. A study in which sewage sludge and wood chips were composted at a ratio of 1:2.5 in Manitoba, Canada, with heaps laid in late autumn, showed that the temperature inside the heap can rise above 65 °C when the ambient temperature is below −20 °C [29]. Other researchers [30] studied low-temperature composting of fresh manure and bedding and found that with an average monthly ambient temperature of −10.8 °C in Southern Alberta, Canada, the center of the heap reached 65 °C. Yet another study [31] showed that winter composting of livestock manure and bedding in passively aerated heaps is possible in southern Idaho, USA, where at ambient temperatures of −15 to −28 °C, thermophilic temperatures were reached within the heap (mean temperature of 50 °C).

To successfully implement cold-weather composting, several points must be considered. Heaps should be laid at favorable atmospheric temperatures. If the laying is carried out in the coldest months of winter, the composting process will start only once the spring–summer period begins [29]. The higher the temperature of the feedstock during heap formation, the better the composting result will be in cold weather [19]. Keeping the number of heap overturns and the intensity of aeration to a minimum [29] allows one to retain warmth and prevent significant drying of the compost mass in winter [18,32]. A further study [32] indicated that low moisture content is a greater limiting factor for composting than low temperature. Additional research [33] recommended that when using natural aeration, piles should be formed at a height of <3 m to ensure an adequate amount of oxygen in the compost mixture.

There is information about the positive effects of inoculating various compost mixture groups with microorganisms in order to enhance the degradation processes and the degree of compost humification [34,35]. In some works [36,37], inoculation led to a beneficial temperature increase in the thermophilic stage of composting. There are other works in which no beneficial effects of inoculation on the composting process were found [38].

The specific objectives of the current study were to determine

- whether it is possible to compost old BWW from bark dumps during the cold period of the year using heaps in an open area;
- how strongly cold weather affects the process of composting old BWW;
- whether microbiological inoculation speeds up the composting process in cold weather conditions.

In order to determine the possibility of implementing BWW field composting during the cold season, an experiment was conducted on an open site for 60 days at ambient air

temperatures from +10 °C to −14 °C (October–December). To assess the negative impact of low atmospheric temperatures on the field composting process, BWW composting was concurrently carried out under controlled laboratory conditions with air temperatures of 30–35 °C along with periodic stirring and hydrating of the compost mixture with tap water.

In the field and laboratory conditions, two compositions of compost mixtures were tested:

- old BWW with the use of mineral fertilizers;
- old BWW with the use of mineral fertilizers and a microbiological inoculum.
- The novelty and practical value of this study is twofold. To begin with, it is the first study to examine the possibility of composting old BWW from bark dumps with the addition of mineral fertilizers in cold weather conditions (up to −14 °C), and furthermore, it is the first to investigate the influence of an inoculum on the process of BWW composting in the aforementioned weather conditions.

## 2. Materials and Methods

### 2.1. Object of Study

The object of study was BWW excavated at an old bark dump near a pulp and paper mill located in the city of Krasnokamsk, in the region of Perm, Russia (Figure 1). More than 1 million tons of waste generated during the debarking process were disposed of at the dump between 1936 and 2005. The total area of the old bark dump is 22.3 ha. The height of the dump varies from 2 to 21 m.

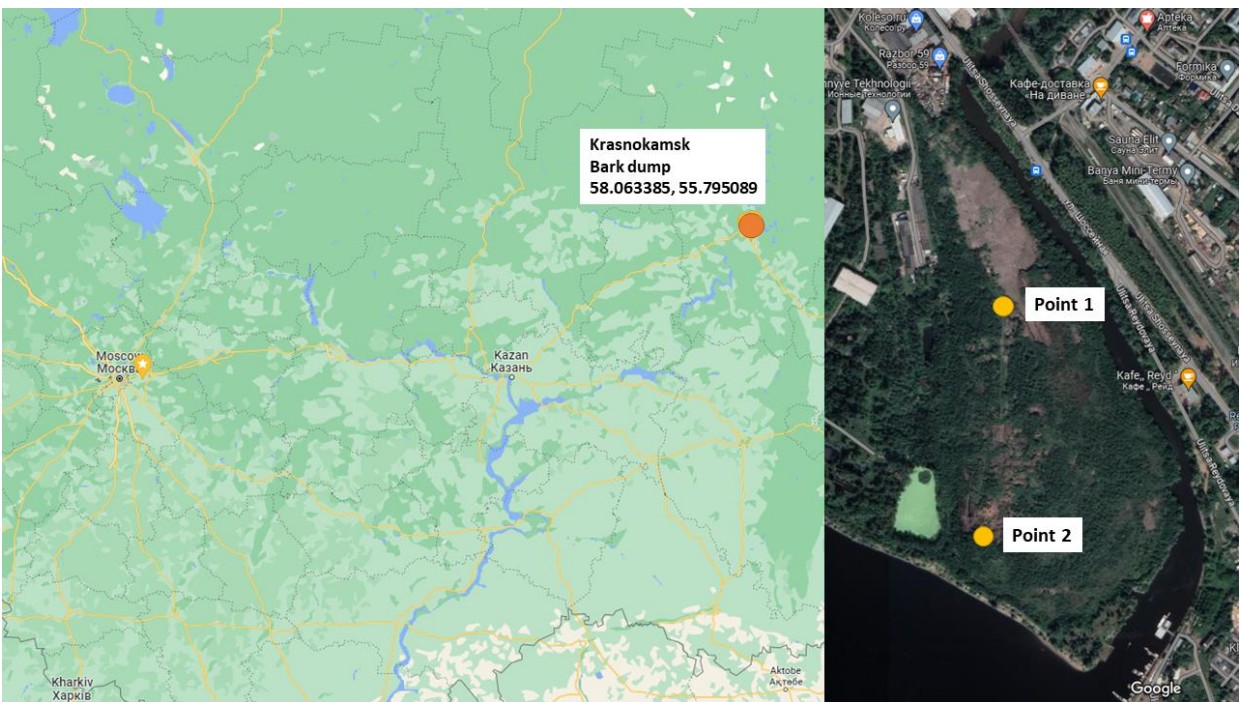

**Figure 1.** Location of the old bark dump in Krasnokamsk and the BWW sampling points.

BWW sampling was carried out from two points (Figure 2a,b) using a bucket excavator with an excavation depth of up to 4 m. When choosing the BWW sampling points, the goal was to collect wastes that characterized various lengths of time spent in the body of the bark dump.

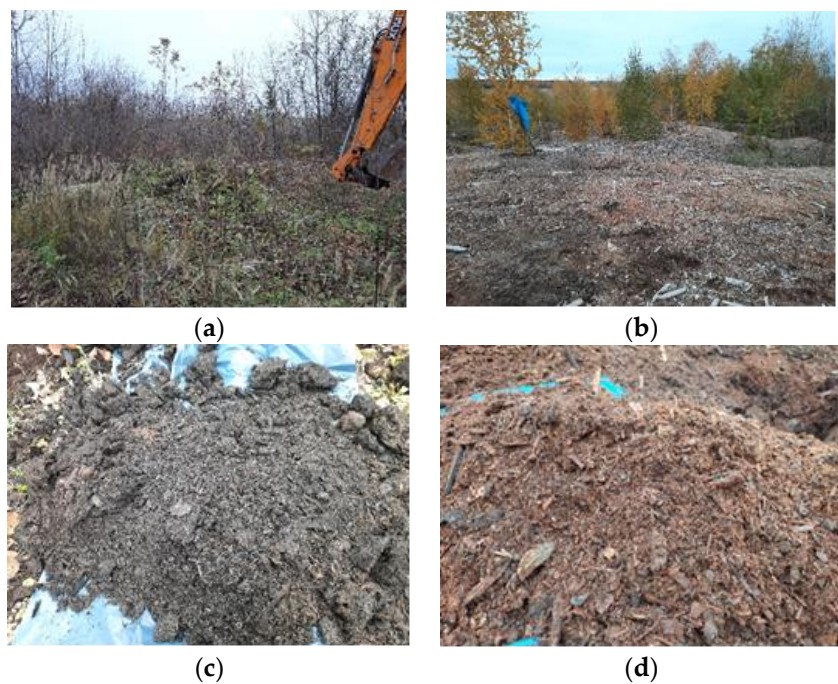

**Figure 2.** BWW sampling process: (**a**) the bark dump surface, sampling point 1; (**b**) the bark dump surface, sampling point 2; (**c**) BWW, sampling point 1; (**d**) BWW, sampling point 2.

### 2.2. Field Composting

In total, 15 tons of BWW from sampling point 1 along with 7.3 tons from sampling point 2 were excavated and delivered to the composting site for the experiment; thus, the ratio of waste masses from the two excavation points was kept close to 2:1.

Mineral fertilizers were added to the BWW at the compost site: urea 0.71 kg/Mg, and diammonium phosphate 0.26 kg/Mg. After the addition of mineral fertilizers, the BWWs at points 1 and point 2 were mixed together with the help of a bucket excavator and divided into two equal heaps. The heaps each had a height of 1.5 m, a base width of 2.7 m, and a bulk density of 700 kg/m$^3$.

Microbiological inoculum was added to heap 1. The commercial microbiological preparation "EMINEXT" was used as the inoculum. It is composed of organic acids, including lactic and benzoic acids; vitamins B and C; amino acids-glycine and tryptophan; and 5 groups of microorganisms in stable symbiosis (photosynthesizing bacteria, lactic acid bacteria, yeast, actinomycetes, fermenting fungi). The choice of a commercial inoculum and the method of its application were based on the manufacturer's recommendations and took into consideration specific features of wood material, the substrate being composted in this case. The inoculum was added during watering from a watering can (15 L/Mg when diluted with tap water to a ratio of 1:10) then the heap had been thoroughly mixed with the help of an excavator.

To maintain the equivalent moisture content in both compost heaps, heap 2 was watered with tap water at a volume equal to that of the introduced inoculum.

The heaps were covered from above with a non-woven polypropylene spunbond fabric (density of 100 g/m$^2$) to prevent swelling of the compost heap and drying out of the upper layer of compost. The covering also helped maintain air exchange during the removal of gases formed in the composting process and ensure the flow of $O_2$ from the outside air to the compost heap.

The preparation of waste for composting and the formation of heaps were carried out at an atmospheric temperature of 10 °C.

Additional wetting, overturning, and mixing of the heaps were not undertaken during field composting as that could have stopped the processes of microbiological destruction

due to heat loss from the compost mixture [29,32]. Watering would also have risked provoking conditions favorable for freezing or icing the entire volume of the heap.

During the field experiment, the temperature and moisture content of the compost mixture in the heaps were measured once every 10 days, while loss on ignition (LOI), respiratory activity ($AT_4$), pH, and chemical oxygen demand (COD) indices were measured once every 20 days.

To determine the potential for using the BWW compost after 60 days of field composting in cold weather, physicochemical and sanitary-hygienic indices were taken based on the Russian standard requirements of GOST R 54534-2011 [39]: moisture, ash content, salt extract pH, Cr, Pb, Cd, Ni, Cu, Zn, As, Hg, COD, biological oxygen demand ($BOD_5$), pathogenic microorganisms incl. *Salmonella*, viable helminth eggs and protozoan cysts, presence of viable larvae and pupae of synanthropic flies, and *Escherichia coli* bacteria.

### 2.3. Old BWW Composting under Laboratory Conditions

In parallel with the field composting, similar compositions of BWW mixtures were composted under laboratory conditions in two open-type polymeric bioreactors with forced mixing for natural aeration. Each 8 L bioreactor was loaded with a compost mixture of BWW weighing 5.5 to 6 kg.

In order to keep moisture at an optimal level during composting, the compost substrate in the bioreactors was periodically moistened with tap water; after each moistening, the entire mixture in the bioreactors was mixed.

The conditions for the laboratory composting process were monitored at intervals of once every three days when it came to the moisture content and pH level of the compost mixture. The LOI, $AT_4$, and COD indices were measured at the end of the experiment after 60 days.

### 2.4. Physicochemical and Microbiological Parameters

The composting process was monitored to track indicators in the compost mixture including temperature, moisture content, acidity (pH), COD, $BOD_5$, LOI, and $AT_4$. These markers directly quantify the conditions and biochemical decomposition process of organic matter in the compost substrate [40].

#### 2.4.1. Temperature

Since the decomposition of organic substances by microorganisms releases thermal energy and heats up the compost mixture [19,41], an increase in the temperature of the compost heap is a clear sign of an active composting process. The opposite is also true in that a decrease in the temperature of the heap is a sign of attenuation in the composting process [42–44]. During the field experiment, the temperature of the compost mixture was measured at three points on each heap using a one-meter multi-zone thermal rod TSHM-A/B, at depths of 7, 50, and 90 cm.

#### 2.4.2. Moisture Content

The optimal moisture level required for microbiological activity during composting is 40–70% [45]. If there is high humidity, anaerobic conditions arise, and the process of decay begins [46,47]. Should there be insufficient moisture, the rate of nutrient exchange between microorganisms and the surrounding substrate decreases [48], as does the rate of microbiological decomposition of organic matter. The moisture content was determined gravimetrically by drying the mixture to a constant weight at 105 °C.

#### 2.4.3. COD, $BOD_5$, pH

During the biodegradation of solid organic matter, intermediate decomposition products are first obtained in the form of organic acids, which decompose into final products in the form of water, mineral salts, and carbon dioxide upon further microbiological destruction [49,50].



COD shows the amount of water-soluble organic substances in water extracted from the compost substrate [40]. This numerical value can be used to estimate how actively the solid organic matter is biodegrading at that moment. According to the Russian standard requirements of GOST R 54534-2011 [39], when COD is less than 700 mg/dm$^3$, the compost can be used as a recultivation material.

BOD$_5$ shows the levels of easily oxidizable, water-soluble organic matter in the aqueous compost extract that can be decomposed by microorganisms. According to the Russian standard requirements of GOST R 54534-2011 [39], when BOD$_5$ is less than 500 mg O$_2$/dm$^3$, the compost can be used as recultivation material.

During composting, the acidity of the compost mixture can vary over a wide pH range from 5 to 8 [51,52], due to the formation and subsequent destruction of intermediate decomposition products in the form of organic acids. The medium acidity level should be kept close to 7–8 in order to be most favorable for the vital activity of microorganisms and the composting process [53].

To determine COD, BOD$_5$, and pH, an aqueous extract was prepared according to the Russian standard requirements of GOST 26483. A dry sample and distilled water in the ratio of 1:2.5 were shaken for 5 min, then the suspension was filtered through an ashless filter paper with a pore diameter of 2–3 μm.

COD was estimated based on ISO 6060:1989 [54] using the oxidation of organic compounds with potassium dichromate in a boiling acid medium, followed by titration of the residual amount with Mohr's salt.

BOD$_5$ was measured according to PND F 14.1:2:3:4.123-97 [55] using the bottle method with iodometric titration.

The pH of the aqueous extract was measured by the potentiometric method using a pH-150MI pH meter.

The pH of the saline extract was measured according to GOST 26483-85 [56]. A dry sample and a solution of 1 n KCl in the ratio of 1:25 were stirred for 1 min. The measurement was carried out in suspension using the potentiometric method on a pH-150MI pH meter.

### 2.4.4. LOI

In the process of composting using biodegradation, the content of solid organic substances decreases, and that of mineral residues increases accordingly [55]. The index of LOI was used to estimate the content of organic solids in the compost mixture. LOI was determined gravimetrically by ignition at 550 °C for 3.5 h with an open crucible, similar to the method described in ASTM-D7348 [57].

### 2.4.5. AT$_4$

The value of microorganism respiratory activity in the substrate is directly proportional to the content of biodegradable organic components to be found there. Thus, it can be used to indirectly estimate a decrease or increase in biodegradable organic matter present in the compost mixture [58,59]. Respiratory activity was assessed in accordance with OENORM S 2027-4:2012: "Evaluation of waste from mechanical and biological treatment. Part 4: Stability parameters—respiratory activity (AT$_4$)" [58]. The measurement was carried out via the manometric method using the OxiTop IS 12 system. The mass of the sample placed in the reaction vessel was 15–20 g and had a moisture level of 60–75%.

### 2.4.6. Metals

In the finished compost, Cr, Pb, Cd, Ni, Cu, Zn, and As content was determined by inductively coupled plasma atomic emission spectrometry (PND F 16.1:2.3:3.11-98 [60]). Hg content was measured by atomic absorption spectroscopy.

### 2.4.7. Sanitary and Hygienic Indicators

The presence of viable larvae and pupae from synanthropic flies was detected based on MU 2.1.7.2657-10 [61]. The substrate was placed in containers covered with calico napkins and left in the laboratory until the flies hatched.

The presence of *Escherichia coli* bacteria and pathogenic microorganisms, including *Salmonella*, was detected and enumerated based on MUK 4.2.3695-21 [62] using the membrane filtration method. This method uses filtration of 5.0–10.0 $cm^3$ of a soil suspension diluted (1:10) through a membrane filter with a pore diameter of 0.45 μm, followed by cultivation of cultures on a differential nutrient medium containing lactose, and finally identification of colonies by cultural and biochemical characteristics.

Detection and estimation of pathogenic microorganisms, including *Salmonella*, was carried out by the method of accumulating pathogenic bacteria in enrichment media (Müller–Kaufmann's medium and magnesium medium), subsequent transfer to dense selective and differential media (bismuth sulphite agar and Ploskirev's bacto-agar), and then the study of biochemical properties of isolated cultures and their serological identification.

Viable helminth eggs and protozoan cysts were determined according to MUK 4.2.3695-21 [62] by direct microscopy. BWW preparation for microscopy of helminth eggs was undertaken according to the method of Romanenko (1996) and for protozoan cysts according to the method of Padchenko (1992).

### 2.5. Statistical Analysis

The data obtained during the experiment were examined using the analysis of variance (ANOVA).

## 3. Results

### 3.1. Properties of Old BWW at the Beginning of Composting

The BWW at point 1 was subject to deeper biochemical destruction in the bark dump body than the BWW at point 2. This conclusion was first made qualitatively by some external indicators and later confirmed via laboratory analyses (Table 1). The black-brown color of the BWW at point 1 and the presence of soil-like substrate mixed with wood inclusions of different sizes indicated an active process of waste humification (Figure 2c), and yet, the smell of wood was present. The BWW at point 2 had the structure of sawdust without visible signs of destruction and a bright red-brown color (Figure 2d), with a strong smell of wood. The results of laboratory analyses showed that the BWW of point 1 was characterized by a 29% lower content of solid organic matter in terms of loss on ignition (LOI) and, respectively, a 68% lower value of respiratory activity ($AT_4$).

**Table 1.** Properties of BWW.

| BWW Sample | LOI (%) | $AT_4$ (mg $O_2$/g) | pH | COD (mg/dm$^3$) |
|---|---|---|---|---|
| Initial compost mix | 76 ± 3 | 3.7 ± 0.4 | 7.6 ± 0.3 | 25,000 ± 4000 |
| BWW, Point 1 | 65 ± 2 | 1.5 ± 0.2 | 7.4 ± 0.1 | 12,500 ± 1110 |
| BWW, Point 2 | 91 ± 3 | 4.7 ± 0.4 | 2.8 ± 0.2 | 6142 ± 250 |

At the BWW excavation point 2, strongly acidic environmental conditions (pH 2.8) were detected. These are detrimental to most microorganisms [40,63] and make the composting process impossible. Therefore, before the start of the experiment, it was decided to combine the BWW masses at point 1 and point 2 at a ratio of 2:1 in the compost mixture so as to have a pH level of 7.6, which is favorable for the vital activity of microorganisms and for greater composting of the waste.

Presumably, the BWW at point 2 contained wood waste that had been in contact with $H_2SO_4$ or $HNO_3$ during the production process at a pulp and paper mill. So, when BWW from the two points was combined, the $H^+$ ions of a mineral acid were used for partial oxidation of available solid organic substrate (lignin, cellulose, hemicel-

lulose, and proteins) in the BWW from point 1. As a result of redox processes, the pH increased to a value of 7.6 and the content of water-soluble organic substances (COD 25,000 mg/dm$^3$), which had resulted from oxidation of solid organic substrate of BWW at point 1, increased significantly.

The second possible reason for the increase in pH was the transition from an anaerobic environment in the bark dump to aerobic conditions during the preparatory stages (excavation, transport, mixing). The presence of oxygen led to more rapid metabolic degradation of the organic acids and thus to an increase in the pH value [64].

### 3.2. Field Composting
#### 3.2.1. Temperature

Table 2 presents the temperature results from the compost mixture in heaps (group mean values, results of the measurements taken at three points on each heap) and atmospheric air measurement taken during the field experiment. Within the entire period of field composting, there were 28 days with an average daily atmospheric temperature above zero; the temperature maximum was 5 °C and the minimum was −14 °C.

ANOVA analysis of variance was performed to determine if there was a statistically significant difference between the average temperatures of the two heaps measured on the same day and at the same depth. As the *p*-value in the ANOVA (Table 3) was in all cases greater than 0.05 (accepted level of significance), it was concluded that the average temperatures between the two heaps did not have statistically significant differences; they were equal to each other. The dynamics of temperature changes in the depth of the two heaps during the field trial also did not differ. The use of the inoculum during field composting did not have a significant effect on the temperature of the process in heap 1.

As the average temperatures in the two heaps were more or less the statistically same, further analysis of the temperature change dynamics was carried out using the general average temperatures (Figure 3).

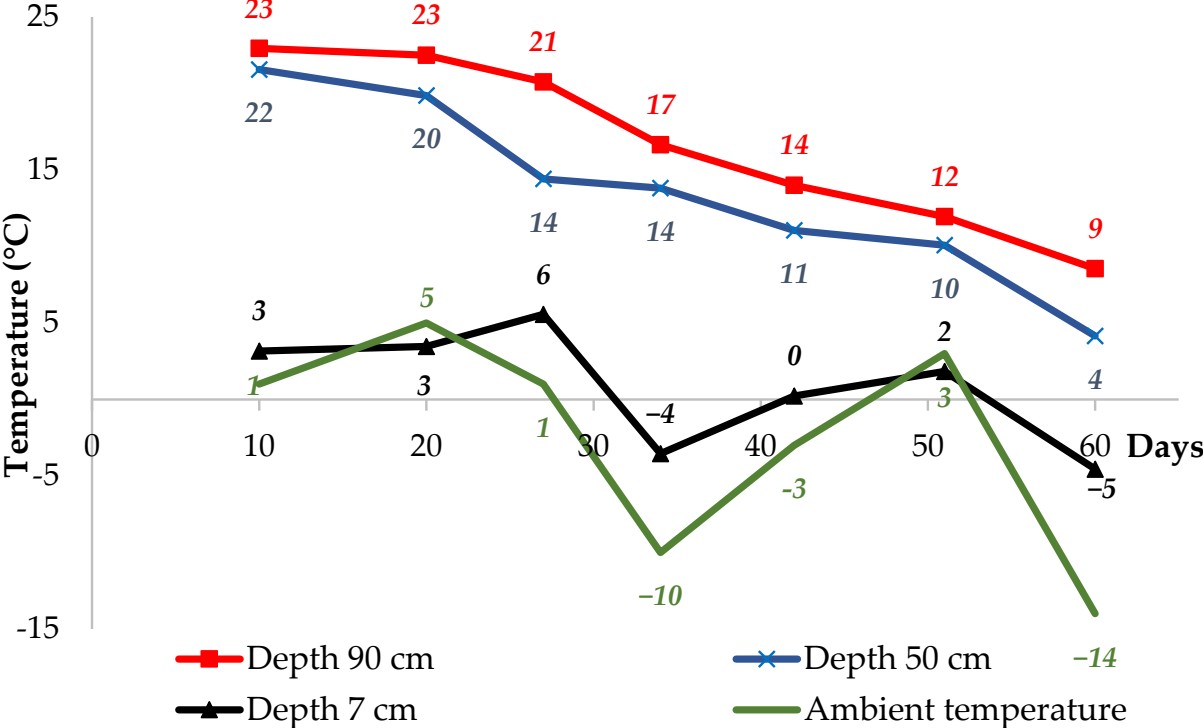

**Figure 3.** Temperature change dynamics in the atmospheric air and in the heaps during field composting.

**Table 2.** Temperature monitoring during old BWW field composting.

| Number of Days | 10 | | 20 | | 27 | | 34 | | 42 | | 51 | | 60 | |
|---|---|---|---|---|---|---|---|---|---|---|---|---|---|---|
| **T (°C) Air** | **+1** | | **+5** | | **+1** | | **−10** | | **−3** | | **+3** | | **−14** | |
| | $\bar{x}$ | SD | $\bar{x}$ | SD | $\bar{x}$ | SD | $\bar{x}$ | SD | $\bar{x}$ | SD | $\bar{x}$ | SD | $\bar{x}$ | SD |
| T (°C) in heap 1 (BWW + fertilizers + inoculum) | | | | | | | | | | | | | | |
| Depth 7 cm | 3.5 | 2.0 | 3.7 | 1.4 | 5.1 | 2.3 | −4.0 | 0.8 | −0.4 | 0.4 | 2.3 | 0.3 | −4.2 | 1.0 |
| Depth 50 cm | 22.4 | 0.7 | 20.2 | 2.3 | 14.3 | 2.3 | 13.3 | 0.4 | 10.4 | 0.5 | 10.0 | 0.5 | 5.0 | 1.7 |
| Depth 90 cm | 23.9 | 2.6 | 23.2 | 1.4 | 21.0 | 2.2 | 16.8 | 0.4 | 14.0 | 0.2 | 12.4 | 0.4 | 9.1 | 1.0 |
| T (°C) in heap 2 (BWW + fertilizers) | | | | | | | | | | | | | | |
| Depth 7 cm | 2.9 | 1.3 | 3.2 | 1.4 | 6.0 | 2.3 | −3.1 | 1.8 | 0.9 | 1.2 | 1.4 | 0.7 | −4.9 | 1.5 |
| Depth 50 cm | 20.8 | 2.7 | 19.6 | 1.5 | 14.5 | 2.8 | 14.4 | 0.96 | 11.2 | 1.5 | 9.2 | 0.9 | 5.3 | 0.8 |
| Depth 90 cm | 22.0 | 3.0 | 22.0 | 3.0 | 20.5 | 2.9 | 16.5 | 2.3 | 14.0 | 2.1 | 11.5 | 1.1 | 7.9 | 1.3 |

**Table 3.** ANOVA results comparing average temperatures of the two heaps.

| Number of Days | Depth of Heap, cm | T °C | | F-Stat | *p*-Value |
|---|---|---|---|---|---|
| | | x̄ | SD | | |
| 10 | 7 | 3 | 2 | 0.2 | 0.68 |
| | 50 | 22 | 2 | 1 | 0.37 |
| | 90 | 23 | 3 | 0.7 | 0.45 |
| 20 | 7 | 3 | 1 | 0.2 | 0.70 |
| | 50 | 20 | 2 | 0.2 | 0.72 |
| | 90 | 23 | 2 | 0.4 | 0.60 |
| 27 | 7 | 6 | 2 | 0.2 | 0.67 |
| | 50 | 14 | 2 | 0.01 | 0.90 |
| | 90 | 21 | 2 | 0.07 | 0.80 |
| 34 | 7 | −4 | 1 | 0.6 | 0.48 |
| | 50 | 14 | 1 | 3.2 | 1.50 |
| | 90 | 17 | 2 | 0.04 | 0.86 |
| 42 | 7 | 0 | 1 | 3.5 | 0.14 |
| | 50 | 11 | 1 | 0.8 | 0.14 |
| | 90 | 14 | 1 | 0.003 | 0.96 |
| 51 | 7 | 2 | 1 | 3.9 | 0.12 |
| | 50 | 10 | 1 | 0.2 | 0.70 |
| | 90 | 12 | 1 | 1.6 | 0.30 |
| 60 | 7 | −5 | 1 | 0.4 | 0.58 |
| | 50 | 4 | 2 | 2.8 | 0.17 |
| | 90 | 9 | 1 | 1.8 | 0.25 |

The influence of atmospheric temperature on the compost mixture temperature was noted to a greater extent in the surface layer of the heaps, at a depth of 7 cm; on the graph, we can observe a synchronous fluctuation in the atmospheric temperature and the temperature of the compost mixture at a depth of 7 cm.

In the deeper layers of the heaps, no obvious temperature jumps were observed in response to external temperature changes. This was established by measurements at depths of 50 and 90 cm. Throughout the entire composting period, the deepest areas of the heaps retained a positive temperature above that of the atmosphere. This, along with the release of heat, was an obvious sign of ongoing microbiological processes as part of organic matter decomposition. On one of the coldest days, with an atmospheric temperature of −14 °C, the heaps froze to a depth of 10–13 cm (Figure 4) on the 60th day of composting.

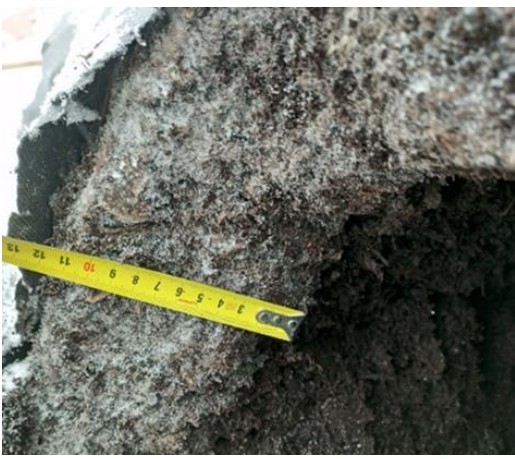

**Figure 4.** The depth of freezing in the heap at atm. air T of −14 °C.

During the first 27 days of field composting, the temperature of the compost mixture inside the heaps (at a depth of more than 50 cm) was kept at 23–20 °C, while the atmospheric temperature ranged from 10 °C to 1 °C. In the days that followed, a uniform decrease in temperature was observed in the heaps at a depth of more than 50 cm to reach values of 4–9 °C with a decrease in the atmospheric temperature down to −14 °C.

The reasons for the uniform decrease in temperature deep inside the heaps starting from the 27th day of the composting could be:

- cooling of the heap due to an average daily decrease in the atmospheric temperature;
- the end of the thermophilic composting stage and the onset of the maturation phase, which is indicated by the equalization of the compost mixture temperature in the heap with the atmospheric temperature [42].

A temperature above 50 °C, as is typical for the thermophilic stage of the composting process [13,65], was not reached deep inside the heaps. Given that the sampled BWW had been stored in the bark dump for more than 10 years under anaerobic conditions, which are typical in any organic waste landfill [66], it was inevitable to see the beginnings of methanogenic decomposition, which is an exothermic process leading to an increase in the landfill mass temperature. Depending on landfill conditions (waste density and amount of precipitation) and composition of the organic waste, the onset of methanogenesis could be any time within the first six years of waste storage and would occur paired with an increase in temperatures up to 35–60 °C [67,68]. Thus, it can be assumed that the thermophilic decomposition phase of the BWW excavated for composting started at BWW point 2 in the bark dump, or that it had already passed to a large extent at BWW point 1 during the period of methanogenesis. The BWW samples at point 1 were mostly affected by biochemical decomposition, as evidenced by their appearance (Figure 2c). The results of analyses (Table 1), in addition to well-developed vegetation and the presence of shrubs on the surface of the bark dump at the point of the BWW sampling, can be seen in Figure 2a. In the presence of methanogenesis, the vegetation would be extremely poor, as illustrated at point 2 (Figure 2b).

The second possible reason for the absence of high temperatures during composting is the lack of rapidly decomposing organic matter in the compost mixture, without which there is no acceleration or prolongation of the thermophilic phase [29], which has temperatures above 55 °C. Wood waste is by its nature a virtually non-decomposable substrate [69,70], and the amount of energy released during its decomposition under aerobic composting conditions may not have compensated for the amount of energy released into the environment in cold weather [29].

### 3.2.2. Compost Indicator Dynamics

Despite the relatively low temperature in the heaps, which is not characteristic of a typical composting process, there was ongoing biochemical destruction taking place in the compost substrate. Composting processes were most active in the first 40 days, with temperatures from 23 °C to 11 °C at the deepest sections of the heap, and atmospheric temperatures ranging from +5 °C to −10 °C. In terms of LOI, there was a 22% decrease in biodegradable solid organic content (Figure 5a). As for COD, there was a notable decrease of 98% in water-soluble organic substance content (Figure 5b). According to the $AT_4$ indicator, the respiratory activity of microorganisms fell by 32% (Figure 5c) due to a decrease in nutrient content available in the organic substrate.

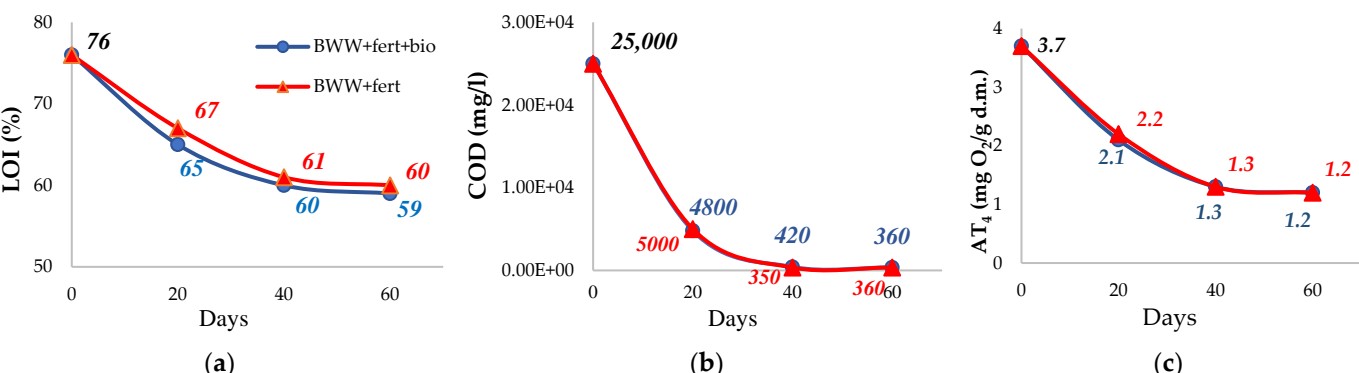

**Figure 5.** Change in the parameters of the compost mixture in heaps during field composting: (**a**) LOI; (**b**) COD; (**c**) $AT_4$.

During field composting, the moisture content and pH of the compost mixture were at optimum levels for the composting process (Table 4).

**Table 4.** Moisture content and pH in heaps during field composting.

| Number of Days | 0 | 20 | 40 | 60 |
|---|---|---|---|---|
| Heap 1 (BWW + fertilizers + inoculum) | | | | |
| pH | $7.6 \pm 0.1$ | $7.7 \pm 0.1$ | $7.9 \pm 0.0$ | $7.6 \pm 0.5$ |
| Moisture content (%) | $65 \pm 1$ | $63 \pm 0$ | $66 \pm 0$ | $67 \pm 1$ |
| Heap 2 (BWW + fertilizers) | | | | |
| pH | $7.6 \pm 0.1$ | $7.5 \pm 0.1$ | $7.9 \pm 0.0$ | $7.7 \pm 2$ |
| Moisture content (%) | $65 \pm 1$ | $60 \pm 2$ | $64 \pm 1$ | $65 \pm 0$ |

*3.3. Laboratory Composting*

Parameters of the compost after 60 days of old BWW composting in the laboratory are shown in Table 5. Furthermore, ANOVA analysis of variance was performed to determine if there was a statistically significant difference between LOI, COD and $AT_4$ values after 60 days of composting in four compost groups:

- BWW + fertilizers + inoculum, field composting;
- BWW + fertilizers + inoculum, laboratory composting;
- BWW + fertilizers, field composting;
- BWW + fertilizers, laboratory composting.

**Table 5.** Parameters of the compost after 60 days of composting.

| | LOI (%) | | COD (mg/L) | | $AT_4$ (mg $O_2$/g) | |
|---|---|---|---|---|---|---|
| | $\bar{x}$ | SD | $\bar{x}$ | SD | $\bar{x}$ | SD |
| Laborotory composting | | | | | | |
| Compost No. 1 (BWW + fertilizers + inoculum) | 57 | 7 | 283 | 66 | 0.9 | 0.1 |
| Compost No. 2(BWW + fertilizers) | 59 | 8 | 300 | 59 | 1 | 0.1 |
| Field composting | | | | | | |
| Heap 1 (BWW + fertilizers + inoculum) | 59 | 5 | 360 | 60 | 1.2 | 0.3 |
| Heap 2 (BWW + fertilizers) | 60 | 9 | 360 | 50 | 1.2 | 0.4 |

Since the *p*-value calculated in the ANOVA (Table 6) for all indicators between the considered compost groups was greater than 0.05 (accepted significance level), it was

concluded that the compost indicator values of all four groups did not have statistically significant differences.

**Table 6.** ANOVA results after 60 days of composting for four compost groups.

|  | **F-Stat** | ***p*-Value** |
|---|---|---|
| LOI | 0.07 | 1 |
| COD | 1.1 | 0.40 |
| $AT_4$ | 2 | 0.18 |

After 60 days of laboratory composting, the result showed that the negative effect of cold temperatures on the process of field composting the BWW with natural aeration was statistically insignificant at the deepest sections of the heaps.

*3.4. Compost Quality*

After sixty days of field composting, the compost mixture of both heaps had acquired blackened wood residues (Figure 6) and lost its wood odor yet retained the smell of forest humus.

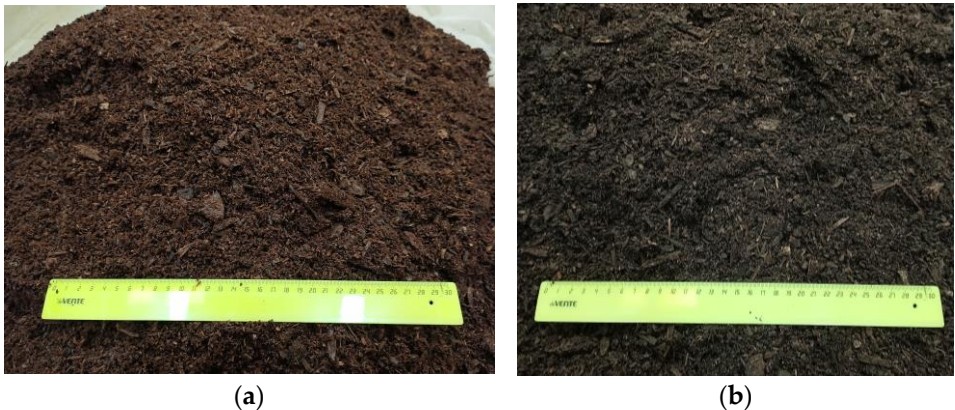

(**a**)         (**b**)

**Figure 6.** BWW appearance: (**a**) at the beginning of composting, (**b**) after 60 days of field composting.

Russian legislation does not provide mandatory requirements for the temperature in heaps during composting to the thermophilic phase. Regulations from the US Environmental Protection Agency do however require that temperatures be above 55 °C to reduce the number of pathogens. Currently, Russia does regulate the quality of the finished compost substrate. The list of controlled indices and their allowable values depends on the intended use: fertilizer, soil, technical ground, or perhaps, soil material for covering landfill or disturbed areas.

Table 7 shows the requisite parameters in order for BWW compost to be used after 60 days of field composting for technical recultivation of disturbed areas.

**Table 7.** Compost quality assessment for use in technical recultivation (GOST R 54534-2011).

| Index | Normative Value | Compost No. 1 BWW + Fert + Bio | Compost No. 2 BWW + Fert | Corresponds to Norm |
|---|---|---|---|---|
| Mass fraction of dry matter (%) | >45 | 38.5 ± 2.7 | 35.4 ± 2.5 | + |
| Ash con tent (%) | >65 (35–65) * | 48 ± 2 | 49 ± 2 | |
| Salt extract pH | 5.0–8.5 | 6.7 ± 0.1 | 6.5 ± 0.1 | + |
| Mercury (mg/kg) | <30 | 0.40 ± 0.02 | 0.43 ± 0.02 | + |
| Chromium (mg/kg) | <2000 | 38 ± 8 | 34 ± 7 | + |
| Plumbum (mg/kg) | <1000 | 33 ± 8 | 28 ± 7 | + |
| Cadmium (mg/kg) | <60 | 0.62 ± 0.31 | 0.46 ± 0.23 | + |
| Nickel (mg/kg) | <800 | 13 ± 5 | 11 ± 4 | + |
| Copper (mg/kg) | <1500 | 240 ± 50 | 210 ± 40 | + |
| Zinc (mg/kg) | <7000 | 190 ± 40 | 140 ± 30 | + |
| Arsenic (mg/kg) | <40 | 1.4 ± 0.7 | 1.6 ± 0.8 | + |
| COD water extract (mg/dm$^3$) | <700 | 360 ± 60 | 360 ± 50 | + |
| BOD$_5$ water extract (mg O$_2$/dm$^3$) | <500 | 170 ± 50 | 160 ± 50 | + |
| Pathogenic microorganisms, incl. *Salmonella* (cells/g) | Absence | Not found | Not found | + |
| Viable helminth eggs and protozoan cysts | Absence | Not found | Not found | + |
| Presence of viable larvae and pupae of synanthropic flies | Absence | Not found | Not found | + |
| *Escherichia coli* bacteria, index | <1000 | 10 | Less than 1 | + |

* Ash content of 35–65% for compost is acceptable, then the compost can only be used when impenetrable base layer and a drainage system are provided.

The assessment showed that the ash content of the composts in both types of heaps (the compost with inoculation—48% and the compost without inoculation—49%) did not reach low enough values for the compost to be used in technical recultivation of disturbed lands. To achieve ash content levels of >65%, it is necessary to continue the composting process, since with greater biochemical decomposition, the content of organic matter will decrease, while that of the mineral salts that make up the ash residue will increase.

In the Russian standard requirements of GOST R 54534-2011 [39], there is a clause related to ash content, which states that ash content of 35–65% is acceptable, but then the compost can only be used when an impenetrable base layer and a drainage system are provided. Thus, the composts from both types of heaps are acceptable for technical purposes such as recovery of disturbed lands, cavities, and excavations formed during open-pit mining. The same composts can also be used for the excavation of minerals, sand, clay, crushed stone development, and for backfilling trenches during the construction and repair of linear structures. All of this is conditioned on using these composts with an impenetrable base layer and a drainage system.

It is worth noting that though the temperature inside the heaps did not rise above 55 °C, pathogenic microorganisms, larvae and pupae of flies, helminth eggs and protozoan cysts were not discovered in the compost, and the presence of bacteria of the *Escherichia coli* group did not exceed the permissible norm. Several reasons could explain this result. Perhaps the BWW was not contaminated to begin with since both the technological process of its formation at the pulp and paper mill, as well as its later storage conditions at the bark dump excluded it from contact with possible sources of pathogenic flora contamination. Another possible reason could be the inactivation of the BWW pathogenic

microflora by elevated temperatures in the bark dump during the exothermic process of methanogenesis, as mentioned above. A third possible explanation is the bactericidal activity of wood metabolites, a topic which is now attracting increasing interest from the scientific community [7,71].

## 4. Conclusions

After field composting previously long-term stored BWW in heaps for 60 days at ambient temperatures of 5 °C to −14 °C, the solid organic matter in the compost mixture decreased by 22% in terms of LOI, the water-soluble organic substances dropped by 98% in terms of COD, and respiratory activity of microorganisms fell by 32% in terms of $AT_4$.

After composting the old BWW in a laboratory for 60 days, the values of the physico-chemical parameters of the compost mixtures were nearly equivalent to those after field composting, which demonstrates that there is an insignificant negative effect of cold atmospheric temperatures on the field composting process.

The use of a microbiological inoculum did not demonstrate high efficiency in contributing to the composting process for old BWW, neither in open composting nor in the laboratory.

According to the Russian standard regulation GOST R 54534-2011 [39], after 60 days of field composting, the compost is deemed suitable for use as the technical ground in the technical recovery of disturbed areas, provided that there is an impenetrable base layer and a drainage system in place.

This study was limited to 60 days of composting and did not cover the entire winter period, which lasts for 120 days with an average minimum temperature of −15 °C in the region of Perm, Russia. It is assumed that with the continuation of the field experiment, the temperature in the heaps would drop to the temperature of the ambient air, and the compost mixture in the heaps could freeze (thereby temporarily suspending the composting process), but with the onset of spring, the composting process would certainly resume.

As BWW is an industrial waste product, there is a risk of the local presence of industrial pollutants in the bark dumpsite, which would slow down the composting process by having a toxic effect on microorganisms. It would also make the finished compost unsuitable for use due to non-compliance with the permissible standards for pollutant content. Consequently, one must carry out input control of the initial BWW composition and output control of the finished compost quality.

Further research should be devoted to studying the composting process over a longer period in winter. It is necessary to assess how possible freezing and thawing of the compost mixture in heaps would affect the composting process. The phytotoxicity and overall quality of the final compost should also be assessed.

The study demonstrated the possibility of field composting old BBW with natural aeration during the cold season. We can, therefore, conclude that this technology is usable all year round, which will, in turn, eliminate the idle time of the composting plant area during the cold season. It will also increase the economic efficiency of the technology and help us more quickly solve the problem of bark dumps as objects that negatively affect the environment.

For some regions of the Russian Federation, such as the Perm region, the possibility of replenishing soil resources with a compost additive is of particular interest, since the predominant soils there are of a soddy-podzolic type, with a thin layer of soil.

**Author Contributions:** Conceptualization, N.S. and Y.M.; methodology, N.S. and Y.K.; validation, Y.M. and A.T.; investigation, Y.M. and A.T.; resources, N.S.; data curation, Y.M. and Y.K.; writing—original draft preparation, Y.M. and Y.K.; writing—review and editing, N.S.; visualization, A.T. and Y.K.; supervision, N.S.; project administration, N.S.; funding acquisition, N.S. All authors have read and agreed to the published version of the manuscript.

**Funding:** The study was performed with financial support from Ministry of science and higher education of the Russian Federation (Project NO FSNM-2020-0024).

**Institutional Review Board Statement:** Not applicable.

**Informed Consent Statement:** Not applicable.

**Data Availability Statement:** Not applicable.

**Conflicts of Interest:** The authors declare no conflict of interest.

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
