# Peer review of "Composting Old Bark and Wood Waste in Cold Weather Conditions"

_sustainability, doi:10.3390/su151410768_

Round 1
Reviewer 1 Report
The manuscript deals with the composting of bark and wood waste in cold climates like in Russia. An interesting comparison was made between the use of microbial inocula to foster biodegradation and composting without inoculation. An additional comparison focused on the ambient air temperature during composting: field experiments in winter were compared with laboratory experiments at higher temperature. The manuscript is easy to read and the concepts written are clear in many parts of the manuscript.
However, the authors must clarify the methodology they adopted to carry out the experiments, since there are some unclear steps that require rearranging the “Materials and Methods” section to improve its clarity. Similarly, the “Results” section needs some explanations about the way the results are presented. “Introduction” and “Conclusions” must be improved too. Please refer to the following comments to improve the manuscript:
Introduction: a clear statement of novelty is missing. Please clearly highlight what novelty the paper brings to the existing literature. In addition, can you explain why you focused on already dumped (and partially biodegraded) BWW instead of starting from fresh BWW? Please motivate your choice in the text.
L15: “LOI” and “AT4” must be defined in the abstract.
L26: the acronym “BWW” must be defined in the main text at its first occurrence.
L31: what about greenhouse gases other than CO2 (e.g., CH4, N2O,…)? Please add some discussion on this topic.
L130: the paper mill has not been mentioned previously in the paper. It’s more appropriate to introduce the paper mill, either at the beginning of section 2 or in the introduction.
L146: “analyzes” should be replaced by “analyses”.
L159-162: nothing has been said about using a mixture in the experiment yet. Is the mixture made exclusively of BWW from samples 1 and 2 in a 2:1 ratio? In this case, how can a mixture of samples with pH 7.4 and 2.8 generate a mixture with higher pH (7.6)? The same applies to COD, whose concentration is exactly twice the concentration of sample 1. Please clarify these points.
L180-182: you should explicitly state that the inoculum was added to the first heap.
L189-190: information on the heaps’ size should be given before in the text.
L194-197: what about aeration? is it natural or forced? Is the specific surface (i.e., ratio between the surface exposed to air and the BWW volume) of the heaps exposed to air the same of the heaps of the field experiment?
L259-261: please indicate what methodologies were used to determine these parameters.
L273: please replace “Table 1” with “Table 2”.
Figure 3: is this graph related to heap 1 or heap 2? Is it an average? If so, why did you consider an average? Please specify.
L347-350: what respect to what heaps in the field experiments? Please be specific.
Figure 6: what do you mean by BWW in the graphs (blue column)? Is it the initial mixture before composting? This is not clear, because in the caption you refer to the situation after 60 days. Please clarify it in the text.
L364: these differences should be visible in Figure 6, but it seems the values reported in the text do not correspond to the values reported in the graphs (especially for COD).
Conclusions: please indicate any weak points or assumptions made during the experiments, the possible future research steps and your point of view on the role and potentiality of composting in this sector.
In my opinion, English language quality is fine. Please check and correct the following minor issues:
L47-51: the main clause is missing.
Figure 6: please replace “conposting” with “composting”.
Author Response
We would like to thank the reviewer for the valuable comments he made. The answers can be found in the attached file

Reviewer 2 Report
Dear Authors,
Please revise the manuscript according to suggestion attached in the text.
Minor revision.
Best regards,

Dear Authors,
Minor revision for english. Please revise the manuscript according to suggestion attached in the text.
Best Regards,
Mahdi Mubarok
Author Response

(The authors gave the same response as above.)

Reviewer 3 Report
Commentaries
The manuscript results are of interest to researchers in environmental sciences, but crucial flaws need to be addressed before to consider to be accepted for publication.
- The authors should start the abstract with an introduction and address the importance of doing this type of composting.
- It is suggested to order the methodology section; it is required to describe the study methodology in the methodology section and not mix it with the results.
- Place subtitles in the methodology for a better understanding
- It is suggested to use atmospheric temperature instead of atmospheric air temperature.
- Perform statistical analyzes (i.e., analysis of variance, such as ANOVA and Tukey's post hoc test if data are normally distributed) to support significant differences mentioned in the manuscript.
Additional commentaries:
Line 2: In the title, it is suggested to eliminate the letter “a” that is found before the word cold
Line 27: Delete the comma after [5] and add ;
Line 31: Join the word “melt water” or add a hyphen between the words
Line 31: Consider adding a transition phrase to improve the flow of your paragraph. For example: As a result, fires that are…
Line 33: Considered adding a comma instead of y (size, quality….)
Line 34: Consider adding a transition phrase to improve the flow of your paragraph. Example: However, since long-term…
Line 43: Change the positive for favorable
Line 50: Change “temperature of the atmospheric air” by atmospheric temperature
Line 54: Add and before rotating drums
Line 64: Add and before organic matter..
Line 68: Consider changing “is the combination of” by “combines”
Line 69: Consider changing “provides protection of” by “protects”
Line 84: It is suggested change “a number of” by several, some, or many
Line 95: Add the hyphen between cold and weather
Line 96: Change the positive for favorable
Line 97: Change “months of winter” by “winter months”
Line 98: Change the preposition with by at
Line 101: Change “warmth” by “warm”
Line 107: Change “in order to enhance the processes of degradation” by “to enhance the degradation processes”
Line 118-128: It is suggested to remove it from the introduction section and place it in the appropriate methodology section.
Line 130: Change the studies by study and were by was
Line 131: Check the wording, it could be better: An object of study was the BWW excavated at the old pulp and paper mill bark dump
Line 132: Delete the letter “s” of the word wastes
Line 133: Change “within the period of” by “from”
Line 144: Change but by However or Nevertheless,…
Line 146. Change the wording, the results of laboratory analyzes… by the laboratory analysis results showed
Line 148: Define the acronym “LOI”, firs time used
Line 149: Define the acronym “AT4”, firs time used
Line 153: Cite figure 2a and 2b in the text
Line 155: Define the acronym “COD”, firs time used, then just put the acronym
Line 155: Instead of a comma, place the units in parentheses
Line 160: Consider changing the verb was by were
Line 161: Add “is” before of favorable
Line 164: Add ; or .
Line 165: Consider changing the verb was by were
Line 167: Consider rephrasing “After addition of……” by “after adding the mineral…”
Line 168: Remove the word point the second time it is mentioned
Line 170: Add “An” after Microbiological inoculant
Line 171: Remove the :
Line 176: Change “taking into consideration” by “considering”
Line 181: Change “the volume equal” by “a volume equal”
Line 183: Add a hyphen or separate the word
Line 189: Change “the height” by “a height”and “the base” by “a base”
Line 191: Consider changing the verb were by was
Line 193: Consider changing the “volume of the heap” by heap volume
Line 197: Consider changing the “the volume” by a volume
Line 211: Consider changing the “the depths of..” by “at 7, 50 and 90 cm depths”
Line 214: Delete “the level of”
Line 217: Delete the letter “s” of the word decreases
Line 227-228: Change “on the basis of” by “based on”
Line 227: Place in subscript the number 3
Line 232: Change “the formation of” by “forming”
Line 241-242: Consider rephrasing “the content of solid organic substances decreases and of” by “solid organic substances content decreases and..”
Line 248: Delete “about”
Line 249-250: Change “in accordance with” by “by, following, per, under”
Line 253: Consider changing the verb were by was
Line 273: Use “cm” instead of meters, as described in the methodology. Also, instead of a comma, place the units of temperature in parentheses
Line 273: Review the numbering of the tables, the table described in line 269 mentions table 2 and line 273 shows number 1
Linea 273: In table 1 (which should be table 2) found in line 273, is the value the average of three replicates?, and the average of the two wells should be obtained with the individual measurements made. Review average data
Line 275 and 277: Change “the depth” by “a depth”, and use “cm” instead of meters
Line 282: Consider changing the verb was by were
Line 292, 294: Change “the depth” by “a depth”
Line 301: Consider changing the word order “composting thermophilic” by “thermophilic composting”
Line 309: Consider changing the word order “to inevitably begin” by “to begin inevitably”
Line 311: Add “s” in the word year
Line 311: Delete “the” in “the waste” and “the increasing”
Line 314: Consider adding Therefore, the BWW….
Line 337: Revie the data of decrease of LOI, COD and AT4, should be the following results: 21.1, 98.6 and 67.6% respectively
Line 350: Place in subscript the number 4
Line 393: Change “used” by “a use”
Line 389, 394: Change “laner” by “a layer”
Line 398: Place the scientific names of microorganisms in italics
Line 407: Place the first letter of Salmonella in capital letters and place in italics, place the meaning of the acronym BOD5 below table 4, and place the scientific names of Escherichia coli in italics
The authors can improve the writing of the manuscript. The manuscript presents grammatical errors, especially in syntax.
Additional commentaries:
Line 2: In the title, it is suggested to eliminate the letter “a” that is found before the word cold
Line 27: Delete the comma after [5] and add ;
Line 31: Join the word “melt water” or add a hyphen between the words
Line 31: Consider adding a transition phrase to improve the flow of your paragraph. For example: As a result, fires that are…
Line 33: Considered adding a comma instead of y (size, quality….)
Line 34: Consider adding a transition phrase to improve the flow of your paragraph. Example: However, since long-term…
Line 43: Change the positive for favorable
Line 50: Change “temperature of the atmospheric air” by atmospheric air temperature
Line 54: Add and before rotating drums
Line 64: Add and before organic matter..
Line 68: Consider changing “is the combination of” by “combines”
Line 69: Consider changing “provides protection of” by “protects”
Line 84: It is suggested change “a number of” by several, some, or many
Line 95: Add the hyphen between cold and weather
Line 96: Change the positive for favorable
Line 97: Change “months of winter” by “winter months”
Line 98: Change the preposition with by at
Line 101: Change “warmth” by “warm”
Line 107: Change “in order to enhance the processes of degradation” by “to enhance the degradation processes”
Line 118-128: It is suggested to remove it from the introduction section and place it in the appropriate methodology section.
Line 130: Change the studies by study and were by was
Line 131: Check the wording, it could be better: An object of study was the BWW excavated at the old pulp and paper mill bark dump
Line 132: Delete the letter “s” of the word wastes
Line 133: Change “within the period of” by “from”
Line 144: Change but by However or Nevertheless,…
Line 146. Change the wording, the results of laboratory analyzes… by the laboratory analysis results showed
Line 148: Define the acronym “LOI”, firs time used
Line 149: Define the acronym “AT4”, firs time used
Line 153: Cite figure 2a and 2b in the text
Line 155: Define the acronym “COD”, firs time used, then just put the acronym
Line 155: Instead of a comma, place the units in parentheses
Line 160: Consider changing the verb was by were
Line 161: Add “is” before of favorable
Line 164: Add ; or .
Line 165: Consider changing the verb was by were
Line 167: Consider rephrasing “After addition of……” by “after adding the mineral…”
Line 168: Remove the word point the second time it is mentioned
Line 170: Add “An” after Microbiological inoculant
Line 171: Remove the :
Line 176: Change “taking into consideration” by “considering”
Line 181: Change “the volume equal” by “a volume equal”
Line 183: Add a hyphen or separate the word
Line 189: Change “the height” by “a height”and “the base” by “a base”
Line 191: Consider changing the verb were by was
Line 193: Consider changing the “volume of the heap” by heap volume
Line 197: Consider changing the “the volume” by a volume
Line 211: Consider changing the “the depths of..” by “at 7, 50 and 90 cm depths”
Line 214: Delete “the level of”
Line 217: Delete the letter “s” of the word decreases
Line 227-228: Change “on the basis of” by “based on”
Line 227: Place in subscript the number 3
Line 232: Change “the formation of” by “forming”
Line 241-242: Consider rephrasing “the content of solid organic substances decreases and of” by “solid organic substances content decreases and..”
Line 248: Delete “about”
Line 249-250: Change “in accordance with” by “by, following, per, under”
Line 253: Consider changing the verb were by was
Line 273: Use “cm” instead of meters, as described in the methodology. Also, instead of a comma, place the units of temperature in parentheses
Line 273: Review the numbering of the tables, the table described in line 269 mentions table 2 and line 273 shows number 1
Linea 273: In table 1 (which should be table 2) found in line 273, is the value the average of three replicates?, and the average of the two wells should be obtained with the individual measurements made. Review average data
Line 275 and 277: Change “the depth” by “a depth”, and use “cm” instead of meters
Line 282: Consider changing the verb was by were
Line 292, 294: Change “the depth” by “a depth”
Line 301: Consider changing the word order “composting thermophilic” by “thermophilic composting”
Line 309: Consider changing the word order “to inevitably begin” by “to begin inevitably”
Line 311: Add “s” in the word year
Line 311: Delete “the” in “the waste” and “the increasing”
Line 314: Consider adding Therefore, the BWW….
Line 337: Revie the data of decrease of LOI, COD and AT4, should be the following results: 21.1, 98.6 and 67.6% respectively
Line 350: Place in subscript the number 4
Line 393: Change “used” by “a use”
Line 389, 394: Change “laner” by “a layer”
Line 398: Place the scientific names of microorganisms in italics
Line 407: Place the first letter of Salmonella in capital letters and place in italics, place the meaning of the acronym BOD5 below table 4, and place the scientific names of Escherichia coli in italics
Author Response

(The authors gave the same response as above.)

Round 2
Reviewer 1 Report
Thank you. The authors improved their manuscript according to the comments raised in the first review round.
Author Response
We would like to thank all the reviewers for their work and valuable comments.
Reviewer 3 Report
The manuscript got better substantially, the authors need to make some corrections.
In general, throughout the manuscript, place the units in parentheses instead of commas. For example, in Figure 5, put LOI (%) or COD (mg/L). In Table 7, for example, Mass fraction of dry matter (%).
Line 22: Create a space between (AT4) and by
Line 236: Put the number 3 in letter
Line 170: Place the object of study in subtopic 2.1
Line 239: Change the title Physicochemical and microbiological parameters
Line 444, 445, 523: Remove °C from first number
The authors can minor editing of English language required
Author Response
We would like to thank all the reviewers for their work and valuable comments.
We have made all the necessary corrections. The text has been proof-read by a native speaker of English.